# Fast High-Resolution Phase Diversity Wavefront Sensing with L-BFGS Algorithm

**DOI:** 10.3390/s23104966

**Published:** 2023-05-22

**Authors:** Haoyuan Zhang, Guohao Ju, Liang Guo, Boqian Xu, Xiaoquan Bai, Fengyi Jiang, Shuyan Xu

**Affiliations:** 1Changchun Institute of Optics, Fine Mechanics and Physics, Chinese Academy of Sciences, Changchun 130033, China; zhanghaoyuan21@mails.ucas.ac.cn (H.Z.); guoliang18@mails.ucas.ac.cn (L.G.); xuboqian@ciomp.ac.cn (B.X.); baixiaoquan@ciomp.ac.cn (X.B.); jiangfengyi@ciomp.ac.cn (F.J.);; 2University of Chinese Academy of Sciences, Beijing 100049, China; 3Chinese Academy of Sciences Key Laboratory of On-Orbit Manufacturing and Integration for Space Optics System, Changchun 130033, China

**Keywords:** active optics, phase diversity, L-BFGS

## Abstract

The presence of manufacture error in large mirrors introduces high-order aberrations, which can severely influence the intensity distribution of point spread function. Therefore, high-resolution phase diversity wavefront sensing is usually needed. However, high-resolution phase diversity wavefront sensing is restricted with the problem of low efficiency and stagnation. This paper proposes a fast high-resolution phase diversity method with limited memory Broyden–Fletcher–Goldfarb–Shanno (L-BFGS) algorithm, which can accurately detect aberrations in the presence of high-order aberrations. An analytical gradient of the objective function for phase-diversity is integrated into the framework of the L-BFGS nonlinear optimization algorithm. L-BFGS algorithm is specifically suitable for high-resolution wavefront sensing where a large phase matrix is optimized. The performance of phase diversity with L-BFGS is compared to other iterative method through simulations and a real experiment. This work contributes to fast high-resolution image-based wavefront sensing with a high robustness.

## 1. Introduction

During the long-term on-orbit observation and operation of space-based large-aperture astronomical telescopes, the influence of space temperature or micro-vibration will gradually cause mirror misalignment and deformation [1,2]. Mirror misalignments and deformations will introduce wavefront aberrations and de-grade the imaging quality of the system. The imaging performance of the space optical system can be maintained by using active optics technology [3,4]. It is necessary to obtain the wavefront information of the optical system, and then actively align the system and correct those aberrations with mirror actuators according to the obtained aberration information.

At present, the commonly used wavefront sensing methods are: Shack-Hartmann sensor [5], pyramid sensor [6], curvature sensing [7] and image-based wavefront sensing method. Image-based wavefront sensing -represents a class of methods that directly utilize image plane intensity measurements to recover the wavefront phase of the pupil plane of an optical system. This class of methods mainly include iterative-transform methods (developed from the Gerchberg-Saxton algorithm) [8,9,10,11,12,13], parametric methods (also known as model-based optimization algorithm or directly called phase diversity algorithm) [14,15,16,17,18], and deep learning methods [19,20,21]. This image-based wavefront sensing method does not require special hardware devices or complex calibration operations, so this type of method is particularly suitable for wavefront sensing in space telescopes [22].

The Phase Diversity (PD) algorithm is a well-known image-based wavefront sensing technique, which usually uses the known defocus aberration between two defocused images to obtain the unknown wavefront aberration of the telescope system [14,15,16]. The PD algorithm does not have high requirements on the hardware, and usually only two defocused images need to be collected when performing wavefront sensing on the optical system [23]. The addition of known diversity phase can improve the efficiency and robustness of the wavefront sensing process, and PD is applicable to extended scene. Since the birth of the PD algorithm, this technology has been widely used in many fields such as adaptive optics, active optics, biological microscopic imaging and quality control of laser beams [17,24,25].

The deviation between theoretical and actual acquired image intensities is evaluated by Fourier optics theory to establish the evaluation function. The key of PD algorithm is to find a suitable optimization algorithm to solve the global optimal value of the evaluation function. Many gradient-based nonlinear optimization algorithms, such as steepest descent (SD) algorithm [26,27], conjugate gradient (CG) algorithm [17] and the quasi-Newton algorithm [28,29], etc. have been applied. Among them, the BFGS algorithm proposed by Broyden et al. is a commonly used quasi-Newton method [30]. They used a matrix that does not contain the second derivative to approximate the Hessian matrix in the Newton method, and solved the problem of finding the second partial derivative in the Newton method. However, each iteration of the BFGS algorithm requires a large amount of storage space. When the optimization problem is large-scale, the storage and calculation of the matrix will be difficult. In response to this problem, Liu and Nocedal et al. proposed a limited-memory BFGS algorithm (L-BFGS), which replaces the previous Hessian matrix by storing a small amount of data from the previous m iterations [31].

Factors such as surface defects of the imaging system will produce high-order aberrations. These high-order aberrations can effectively influence the intensity distribution of the point spread function (PSF) which will decrease the accuracy of wavefront sensing. Therefore, high-resolution phase-diversity wavefront sensing is usually required to accurately detect aberrations in optical systems. However, high-resolution phase diversity wavefront sensing is restricted with the problem of low efficiency and stagnation, this paper proposes a fast high-resolution PD wavefront sensing with L-BFGS algorithm. The algorithm takes all pixels on the phase plane as unknown quantities to solve the problem, and uses the analytical gradient calculation method proposed by Fienup et al. to improve the computational efficiency of the algorithm [12].

The remainder of this paper is organized as follows. In Section 2, we review the classical PD algorithm and the L-BFGS algorithm, and propose the basic flow of the fast high-resolution PD algorithm. Section 3 is the simulation analysis of algorithm performance. Section 4 is the experimental verification part. And in Section 5, we conclude this paper.

## 2. Principle of Fast High-Resolution Phase Diversity Wavefront Sensing with L-BFGS Algorithm

### 2.1. Review of Phase Diversity Algorithm

In this section, we will give a brief description of the principle of the PD algorithm [14,15,16]. For a diffraction-limited incoherent imaging system, the image at the focal plane of the system is the convolution of the object with the system PSF:(1)$$d_{1}(r)=o(r)*h_{1}(r)+n_{1}(r),$$
where, $d_{1}(r)$ is the light intensity distribution at the focal plane of the system, o(r) is the unknown target, $h_{1}(r)$ is the PSF at focal plane, $n_{1}(r)$ is the detector noise (Gaussian distribution), ∗ represents the convolution operation, r is the two-dimensional position vector of the image plane, and:(2)$$h_{1}(r)={\left|F\left\{A(\rho )\cdot e^{j\varphi (\rho )}\right\}\right|}^{2},$$
where, A(ρ) is the pupil amplitude, φ(ρ) is the unknown wave aberration of the system, ρ is the two-dimensional position vector of the pupil plane, $F\left\{\cdot \right\}$ represents the Fourier transform operation.

The image and PSF of the defocused plane of the system can be expressed as:(3)$$d_{2}(r)=o(r)*h_{2}(r)+n_{2}(r),$$
(4)$$h_{2}(r)={\left|F\left\{A(\rho )e^{j[\varphi (\rho )+\Delta (\rho )]}\right\}\right|}^{2},$$
where, $d_{2}(r)$ is the light intensity distribution at the defocus plane of the system, $h_{2}(r)$ is the PSF at defocus plane, $n_{2}(r)$ is the detector noise, and Δ(ρ) is the known defocus aberration.

We usually use Zernike polynomials to represent the wavefront aberrations of optical systems [32]:(5)$$\varphi (\rho )=\sum_{i=4}^{N}C_{i}Z_{i}(\rho ),$$
where, $Z_{i}(\rho )$ is the i-th term of the Zernike polynomial, and $C_{i}$ is the coefficient of the i-th term of the Zernike polynomial. Therefore, given a set of coefficients $a=\left[C_{1},C_{2},C_{3},\ldots ,C_{N}\right]$, the corresponding system wavefront aberration can be obtained.

Then, the evaluation function is constructed using the maximum likelihood estimation method:(6)$$E=\sum_{k=1}^{2}\sum_{r}{\left[d_{k}(r)-o(r)*h_{k}(r)\right]}^{2},$$

If the unknown system wavefront aberration is to be obtained, it is necessary to find the global optimal solution that makes the evaluation function shown in formula (6) the minimum value. At this time, the problem of using the PD algorithm to solve the system wave aberration is transformed into a nonlinear optimization problem: the unknown wavefront phase information can be obtained by selecting an appropriate optimization algorithm. 

### 2.2. The Principle of L-BFGS Algorithm and the Application of Analytic Gradient in L-BFGS Algorithm

Compared with the BFGS algorithm, the L-BFGS algorithm reduces the requirement for storage capacity, it avoids the calculation of large-scale matrix and improves the calculation efficiency. The iterative formula of the L-BFGS algorithm is as follows:(7)$$v_{k+1}=v_{k}+\alpha_{k}H_{k}g_{k},$$
where, $v_{k}$ and $v_{k+1}$ are the iteration results of the kth and k+1th iterations respectively, $\alpha_{k}$ is the step size of the kth iteration, $g_{k}$ is the gradient of the kth iteration, and $H_{k}$ is the Hessian matrix (a matrix of 2nd order partial derivatives) of the kth iteration. Define $V_{k}=I-\rho_{k}y_{k}s_{k}^{T}$, $\rho_{k}=\frac{1}{y_{k}^{T}s_{k}}$, $y_{k}=g_{k+1}-g_{k}$, $s_{k}=v_{k+1}-v_{k}$. Then the $H_{k}$ can be expressed as:(8)$$H_{k+1}=V_{k}^{T}H_{k}V_{k}+\rho_{k}s_{k}s_{k}^{T},$$

At the same time, $H_{k}$ can be obtained by using the initial positive-definite matrix $H_{0}=I$ and the information in the previous m steps. Therefore, Equation (8) can be expressed by the initial positive-definite matrix:(9)$$\begin{array}{l} H_{k+1}=V_{k}^{T}\ldots V_{k-m}^{T}H_{0}^{}\ldots V_{k}\\ +\rho_{k-m}(V_{k}^{T}\ldots V_{k-m+1}^{T})s_{k-m}s_{k-m}^{T}(V_{k-m+1}^{T}\ldots V_{k}^{T})\\ +\rho_{k-m+1}(V_{k}^{T}\ldots V_{k-m+2}^{T})s_{k-m+1}s_{k-m+1}^{T}(V_{k-m+2}^{T}\ldots V_{k}^{T})\\ +\ldots \\ +\rho_{k+1}V_{k}^{T}s_{k+1}s_{k+1}^{T}\\ +\rho_{k}s_{k}s_{k}^{T} \end{array}$$

In this paper, the whole phase plane is taken as the calculation target, and the gradient information of each pixel value on the phase plane is solved separately. Fienup et al. proposed an analytic gradient expression based on the Fourier transform [12], which can be used to obtain the derivative of Equation (6) for the phase plane:(10)∂E∂φ=−2ImP(φ)F−1Gw∗(φ)
where, $G^{w^{*}}(\varphi )=2\cdot \left[{\left|F(\varphi )\right|}^{2}-i\right]\cdot F{\left(\varphi \right)}^{*}$, $F(\varphi )=F\left\{A(\rho )e^{j\varphi (\rho )}\right\}$, the superscript ∗ represents complex conjugate, $F^{-1}\left\{\cdot \right\}$ represents the inverse Fourier transform, Im⋅ represents the imaginary part of a complex number, and the generalized pupil function $P(\varphi )=A(\rho )e^{j\varphi (\rho )}$.

Equation (10) shows that only one inverse Fourier transform is needed to obtain the gradient information of n⋅n pixel values on the phase plane, and the complexity of calculating the gradient is reduced from O(n⋅n) to O(1). And Equation (10) corresponds to $g_{k}$ in Equation (7), the application of this analytical gradient expression in the L-BFGS algorithm can significantly improve the convergence efficiency of the algorithm.

### 2.3. Fast High-Resolution Phase Diversity Wavefront Sensing with L-BFGS Algorithm

Different from the traditional PD algorithm that takes the Zernike coefficient as the solution target, the fast high-resolution PD (hereinafter referred to as high-resolution PD) algorithm proposed in this paper uses all the pixels on the phase plane as the unknown quantity to solve the problem. Combining the analytical gradient calculation method shown in Formula (10) with the L-BFGS algorithm, the computational complexity of the Hessian matrix will change from O(n⋅n) to O(n⋅m) (usually m is much smaller than n), thus improving the computational efficiency. The algorithm process is as Figure 1:

## 3. Simulations

### 3.1. System Parameter Setting

We set the diameter of the primary mirror of the optical system to be 2 m, the focal length of the system to be 28 m, the observation wavelength to be 625 nm, the number of CCD samples to be 256 × 256 pixels, the pixel size of the CCD to be 5.5 µm, and the defocus distance between the two images to be 10 mm.

### 3.2. Solution Accuracy Analysis of High-Resolution PD Algorithm

In this paper, root mean square (RMS) is used to represent the magnitude of wavefront aberration, and use root-mean-square error (RMSE) to measure the accuracy of the solution. Three sets of Zernike polynomial coefficients with different RMS are used to simulate the wavefront aberration, and the generated PSF image is substituted into the high-resolution PD algorithm to solve the wavefront aberration. Then, high-order aberrations are added for simulation to analyze the ability of the high-resolution PD algorithm to solve wavefront aberrations when high-order aberrations are included. The calculation result is shown in Figure 2:

The following conclusions can be drawn from Figure 2:The high-resolution PD algorithm achieves convergence when introducing different high-order aberrations;The calculation accuracy of wavefront aberrations is affected by higher order aberrations;The larger the wavefront aberration, the lower the solution accuracy.

### 3.3. Comparative Analysis

#### 3.3.1. Comparative Analysis of Convergence Efficiency

When high-order aberrations are involved, the high-resolution PD algorithm, the traditional PD algorithm for solving Zernike coefficients, and the GS algorithm are used to solve the aberrations to compare the convergence efficiency of the three algorithms. The results are shown in Figure 3:

The following conclusions can be drawn from Figure 3:The high-resolution PD algorithm converges faster and has higher solution accuracy than the traditional PD algorithm;The GS algorithm guarantees the solution accuracy through multiple cross iterations but greatly sacrifices the convergence efficiency;The high-resolution PD algorithm can also quickly converge while ensuring the solution accuracy. We can see the proposed algorithm converges 2 times faster than the GS algorithm, which is commonly used now.

#### 3.3.2. Comparing Analysis of Solution Accuracy

Three sets of Zernike coefficients are selected to simulate wavefront aberrations with higher order aberrations. The results of the three algorithms are shown in Figure 4:

The following conclusions can be drawn from Figure 4:In the case of high-order aberrations in the system, the accuracy of the high-resolution PD algorithm for solving aberrations is better than the other two algorithms;When the wavefront aberration is small, the GS algorithm can correctly solve the aberration. However, when the wavefront aberration is large, the GS algorithm falls into the local extremum, and the correct aberration information cannot be obtained;In the case of high-order aberrations in the system, the traditional PD algorithm cannot solve the aberrations;The solution accuracy decreases with the increase of wavefront aberration.

#### 3.3.3. Comparing Analysis of Robustness Analysis

In order to be more realistic, the noise related to the intensity and satisfying the Gaussian distribution is added to the simulated PSF image. Use Peak Signal-to-Noise Ratio (PSNR) as a criterion for evaluating noise magnitude:(11)$$PSNR=20log_{10}\left(\frac{S_{peak}}{\sqrt{S_{peak}+\sigma_{read}^{2}+\sigma_{dark}^{2}}}\right)$$
where, $S_{peak}$ is the maximum value of the intensity in the noise-free image, $\sigma_{read}^{2}$ and $\sigma_{dark}^{2}$ are the variance of read noise and dark current noise, respectively.

In order to further verify the effectiveness of the proposed method, 100 groups of Monte Carlo tests are carried out and the results are presented in Figure 5. In the range of $\left[-0.3\lambda ,0.3\lambda \right]$, 100 sets of Zernike coefficients are randomly generated to verify the robustness of the algorithm under different SNR conditions:

The following conclusions can be drawn from Figure 5:In the case of low signal-to-noise ratio, the high-resolution PD algorithm and GS algorithm are relatively stable, while the traditional PD algorithm is more likely to fall into local extremum;When the noise becomes larger, the high-resolution PD algorithm can still maintain stability, while the stability of the GS algorithm will gradually decrease;

## 4. Experimental

The experimental verification optical path shown in Figure 6 was designed and built by using the existing equipment in the laboratory. Which is mainly composed of a telescope system, an interferometer, a CCD camera, and a beam splitter prism. And a flat mirror is placed in front of the main mirror of the telescope to form a self-collimating optical path. In order to reduce the influence of airflow disturbance, an interferometer and a CCD camera are used to collect data simultaneously.

A set of images containing known defocus aberrations is collected by moving the CCD camera along the optical axis, and then the high-resolution PD algorithm is used to solve the wavefront aberrations in the optical system. At the same time, the high-resolution algorithm is verified with the data collected by the interferometer. The result is shown in Figure 7:

In order to avoid the influence of air flow disturbance on the experimental results, four sets of images were collected in each of the two states. Subsequently, the solution results are expressed by Zernike polynomials, and the solution results of the 4th to 7th items of the Zernike polynomials are shown in Table 1:

It can be seen from Figure 7 and Table 1 that under different conditions, the high-resolution PD algorithm can successfully solve the wavefront aberration of the optical system, and the result is relatively stable.

## 5. Conclusions

High-resolution phase diversity wavefront sensing is of great importance in the area of optics. However, high-resolution phase diversity wavefront sensing is restricted with the problem of low efficiency and stagnation. We propose a fast and high-resolution PD wavefront sensing method based on the L-BFGS algorithm to address this problem. The algorithm takes all pixels on the phase plane as unknown quantities, and uses the analytical gradient shown in Equation (10) to improve the computational efficiency of the algorithm.

Simulations are performed to demonstrate the accuracy and convergency of the proposed algorithm. On one hand, it is shown that the proposed algorithm can accurately recover high-resolution wavefront phase over a large range of wavefront error (i.e., this algorithm is robust to stagnation problem). On the other hand, the results also show that this algorithm is superior over other algorithms in accuracy and convergence efficiency.

Real experiments are performed to further validate the effectiveness of the proposed method. It is shown that the proposed method can accurately recover high-resolution wavefront phase when two defocused images are available. 

This work provides a feasible solution to the problem of low efficiency and stagnation in high-resolution phase diversity wavefront sensing.

## Figures and Tables

**Figure 1 sensors-23-04966-f001:**
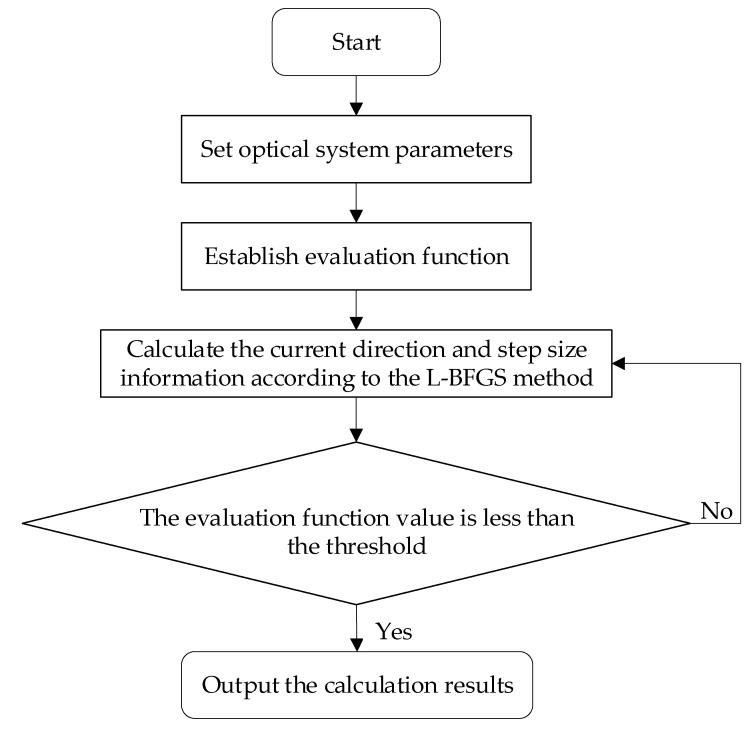
Algorithm flowchart.

**Figure 2 sensors-23-04966-f002:**
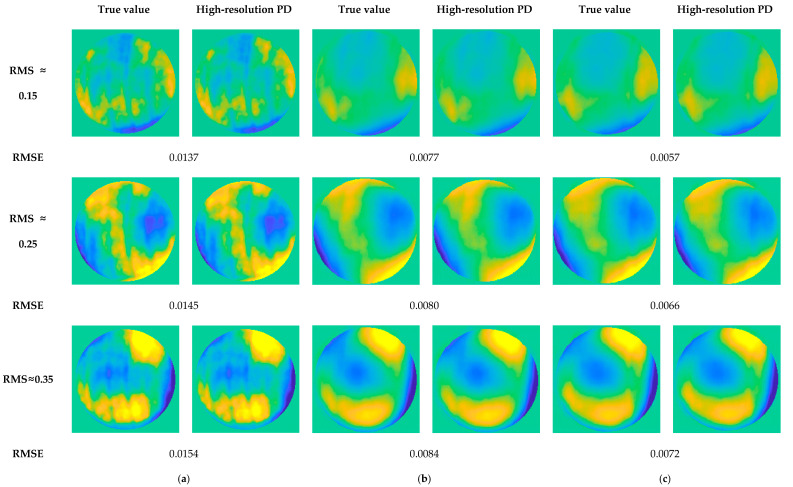
The simulation analysis results of solution accuracy of high-resolution PD algorithm. (**a**–**c**) are in three different high-order aberration situations, respectively. In each case, the simulation analysis is carried out by adding different magnitudes of low-order wavefront aberrations.

**Figure 3 sensors-23-04966-f003:**
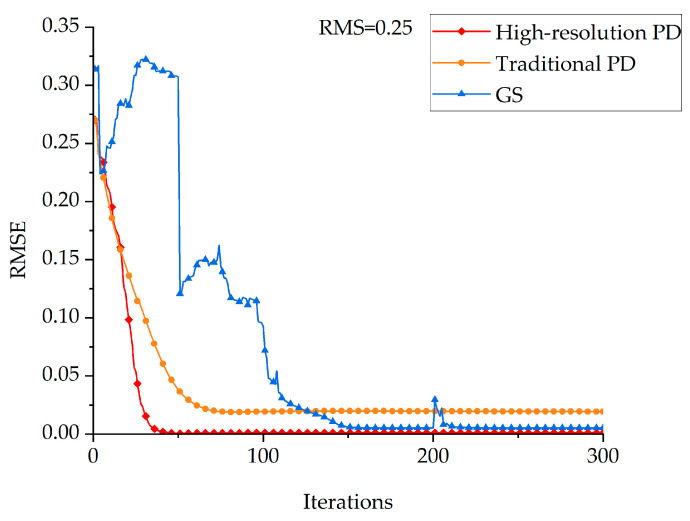
The solution results of the three algorithms for wavefront aberrations when the wavefront aberration size is RMS = 0.25 and with high-order aberrations.

**Figure 4 sensors-23-04966-f004:**
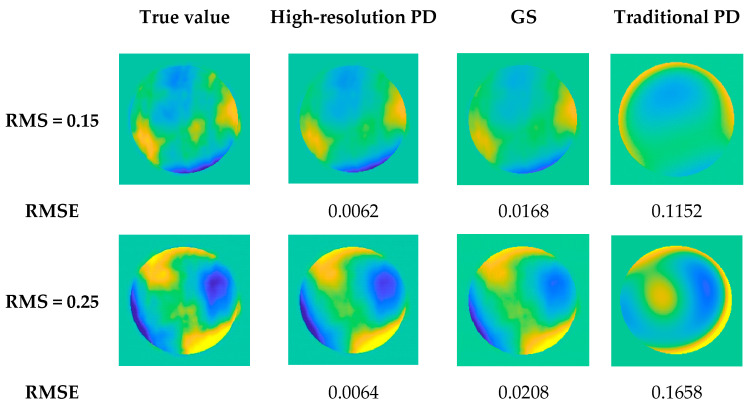
The result of the solution of the three algorithms.

**Figure 5 sensors-23-04966-f005:**
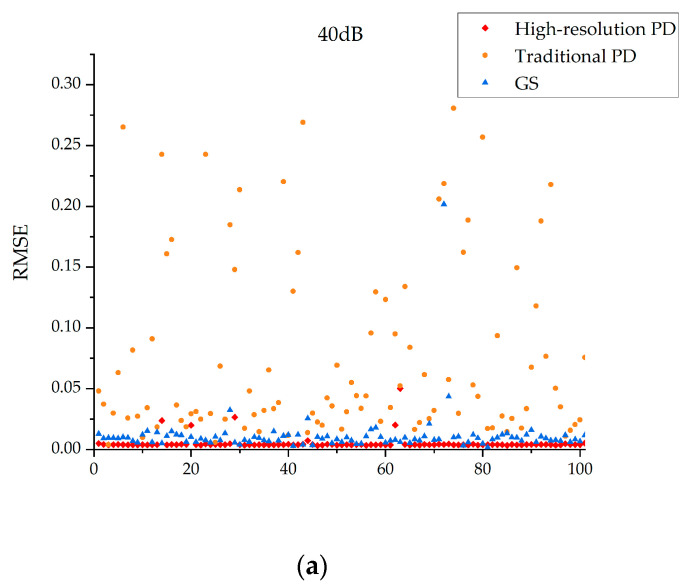
The algorithm robustness analysis under different SNR conditions. The SNRs of (**a**–**c**) are 40 dB, 30 dB and 20 dB respectively.

**Figure 6 sensors-23-04966-f006:**
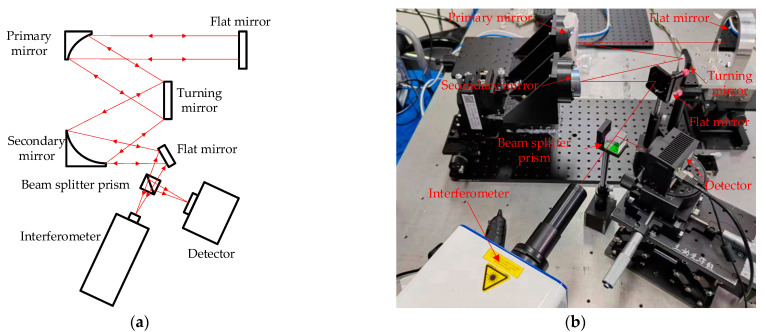
High-resolution PD algorithm experimental optical path diagram, (**a**) is the schematic diagram, and (**b**) is the experimental optical path diagram. The light emitted by the interferometer is irradiated to the flat mirror after passing through the secondary mirror, the turning mirror and the primary mirror of the telescope, and then returns according to the original path. The returned light is divided into two paths by the beam splitter prism, one path of light returns to the interferometer and the other path of light converges on the focal plane of the detector.

**Figure 7 sensors-23-04966-f007:**
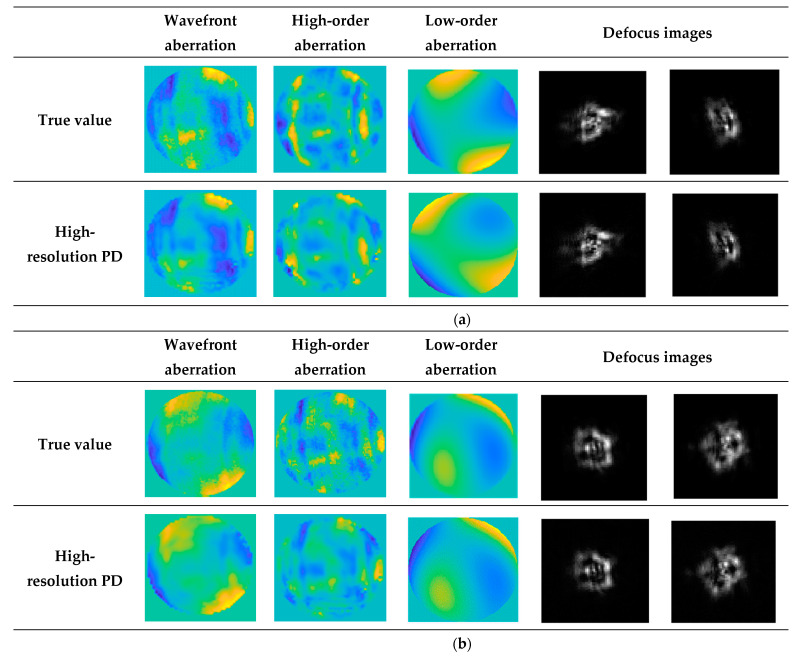
Experimental results. (**a**,**b**) are the experimental results conducted in two states of the optical system, respectively This figure shows the results of the high-resolution PD algorithm restoration of wavefront aberrations, higher-order aberrations, lower-order aberrations, and defocused images. It can be seen that the wavefront aberrations of the optical system are successfully solved by the high-resolution PD algorithm.

**Table 1 sensors-23-04966-t001:** Experimental verification results of high-resolution PD algorithm. (a) and (b) represent the two states of the optical system, respectively. A–D represents four sets of data collected in the same state.

(a)		**C4**	**C5**	**C6**	**C7**	**RMSE**
	True value	−0.0090	−0.1847	0.0539	0.0699	
	A	0.0069	−0.1792	0.0463	0.0902	0.0103
	B	0.0204	−0.1675	0.0490	0.0726	0.0140
	C	0.0184	−0.1706	0.0560	0.0687	0.0126
	D	0.0165	−0.2076	0.0423	0.0792	0.0149
(b)		**C4**	**C5**	**C6**	**C7**	**RMSE**
	True value	−0.0526	0.0636	0.0837	0.0306	
	A	−0.0406	0.0785	0.0775	0.0285	0.0081
	B	−0.0438	0.0787	0.0553	0.0199	0.0129
	C	−0.0453	0.0814	0.0787	0.0200	0.0089
	D	−0.0688	0.0841	0.0558	0.0215	0.0149

## Data Availability

Not applicable.

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
