# Peer review of "Fast High-Resolution Phase Diversity Wavefront Sensing with L-BFGS Algorithm"

_sensors, 2023, doi:10.3390/s23104966_

Round 1
Reviewer 1 Report
The authors propose a fast and high-resolution phase diversity wavefront sensing base on image processing with increased robustness. The subject is of interest and within the scope of Sensors Journal.
The paper is structured and in general well organized and written. There are some minor redaction bugs and typos. The abstract is descriptive and the introduction give an appropriate background and perspective. The figures are illustrative and representative, although fig. 4 could be improved. Authors describe the wavefront phase using Zernike polynomials including high order aberrations, and employ a modified BFGS algorithm to reduce storage requirements and avoid large matrix calculations. The mathematical development and the theoretically basis are founded, although somewhat basic. Some references are outdated (1, 2,12, 13, 14, 20) and although they have a context importance, authors should include extra references to recent works. The way to refer to equations is unusual but understandable.
Conclusion section resembles a re-statement of the research; it is very similar to the abstract. This section must remark the results and their relevance, in addition to propose areas of improvement
Author Response
Dear Editor,
Thank you for your letter and for the reviewers’ comments concerning our manuscript entitled “Fast High-resolution Phase Diversity Wavefront Sensing with L-BFGS Algorithm”. We are deeply grateful that our paper was reviewed by those who are the real expert in this field. All of the comments are really valuable and helpful for revising and improving our paper, as well as the important guiding significance to our future researches. We have studied the comments carefully and made corresponding revisions according to their advices. The detailed content is shown in the attachment.
Best regards,
< Haoyuan Zhang, Guohao Ju > et al.

Reviewer 2 Report
This paper proposes a fast high-resolution PD wavefront sensing with L-BFGS algorithm. The algorithm takes all pixels on the phase plane as unknown quantities to solve the problem, and uses the analytical gradient calculation method to improve the computational efficiency of the algorithm.
This manuscript is adequate and the text is basically well-written. It can be published in Sensors after minor revision. I think the author should explain the following issues as support:
1. The abbreviation L-BFGS should be explained in the introduction, otherwise readers will only understand its meaning after reading line 67.
2. Please explain in lines 60 to 61 why “the quasi-Newton algorithm” is abbreviated as BFGS?
3. In lines 72 and 89, the PSF is duplicated.
4. Please explain line 128, what is the computational complexity O (n * n)? How to define it?
5. Suggest drawing the algorithm flow from lines 148 to 156 as a flowchart.
6. In the simulation of 3.2, please provide the coefficients of the Zernike polynomial.
7. The size of the wavefront phase map in Figures 1, 3, and 6 is different. Does this have an impact on RMSE? Please explain why the size of the wavefront phase map in Figures 1, 3, and 6 is different.
8. The text description in Figure 5 does not match the annotations on the figure, such as“beam splitting prism” and “beam splitter prism”.
Author Response

(The authors gave the same response as above.)

Reviewer 3 Report
Dear Authors,
The paper under consideration presents application of a new algorithm for a wavefront measurements. The topic is within the scope of Sensors journal and could be of some interest for the readers. The sensing principle is demonstrated in a simple experiment, which prooves the advantages of the proposed algorithm.
However, I recommend to pay attention to the following points before accepting the paper for publication:
Major points:
1. In the abstract it is stated that the wavefront sensing method will be used for measurement and compensation of the manufacturing errors, while in the introduction the misalignmens appearing during the flight of a spaceborne instrument are discussed. So, the question is - what is the target and reachable frequency of the wavefront error in the time domain? Could one measure only static or slo-periodic WFE? If yes, this is still a good measurement technique , but not actually "sensing" applicable for active and adaptive optics systems. If no, then what is the minimum measurable period of the WF evolution? I believe that it's a critical point since the algorithm is time-consuming (in comparison with the sensors operating in the pupil plane).
2. Section 2.2 - it is not clear how the fact that the measurements are repaeted in 2 focal positions is reflected in the equations.
3. Section 3.1 and further - what is a rationale behind the parameters used in the testcase? E.g. the focal length seems to be too large for a space instrument.
4.The simulations and experiment use monochromatic light, but for a real in-flight observations the sensor would have to deal with a polychromatic image. What would be the effect of this change?
5. Sec 3.2 and 3.3 each time some conclusions are made about the algorithms accurac, convergence etc, it wuld be very useful to have a numerical estimation like "the proposed algorithm converges 1.5 times faster than the one, which is commonly used now".
6. As the seprate conclusions will be revised I suggest to extend the "5.Conclusions" section - provide some key values and make clear statement as a list of advantages.
Minor points:
1. Abstract and throughpot the text - L-BFGS stangs for limited memory Broyden–Fletcher–Goldfarb–Shanno algorithm and the acronym should be explained on the first use. Also it is not recommended to use the acronym in the abstract.
2. Line 38 - it is not correct to use the WFS acronym only for the image-based WFS.
3. Line 54 - the classification is not correct, since it mixes division bty frequencies (active/adaptive) and by application (biomedical/astronomica/laser)
4. Line 121 - it could be useful to remind the reader here that the Hessian is a martix of 2nd order partial derivatives.
5. Fig.1 - It would be nice to explain the RMS/RMSE acronyms on the first use.
6. Fig.2 - the plot would be easier to read with markers of different shapes.
7. Sec 3.3.3. - "Comparative analysis"?
8. Line224 - the conclusion that the accuracy decreases with the raise on noise is really trivial.
9. Table 1 - why these specific modes are used?
In general, I believe that the paper can be published in Sensors, but requires some moderate revisions.
Author Response

(The authors gave the same response as above.)

Reviewer 4 Report
I would recommend improve some parts of the presented article.
1. Page 5 you pointed 'The larger the wavefront aberration, the lower the solution accuracy.' What do you mean, the impact of the high-order aberaations, low-order aberrations or overall aberrations?
2. Could you add numerical values of the initial/residual aberrations for simulations and experimental results?
After answering on these questions the paper could be published in Sensors.
Author Response

(The authors gave the same response as above.)
